# Biochar Production and Demineralization Characteristics of Food Waste for Fuel Conversion

**DOI:** 10.3390/molecules28166114

**Published:** 2023-08-17

**Authors:** Kwang-Ho Ahn, Dong-Chul Shin, Ye-Eun Lee, Yoonah Jeong, Jinhong Jung, I-Tae Kim

**Affiliations:** 1Department of Environmental Research, Korea Institute of Civil Engineering and Building Technology, 283 Goyang-daero, Ilsanseo-gu, Goyang-si 10223, Republic of Korea; khahn@kict.re.kr (K.-H.A.); yeeunlee@kict.re.kr (Y.-E.L.); yoonahjeong@kict.re.kr (Y.J.); jinhong98@kict.re.kr (J.J.); 2Department of Smart Construction and Environmental Engineering, Daejin University, 1007 Hoguk-ro, Pocheon-si 11159, Republic of Korea; dcshin@daejin.ac.kr

**Keywords:** food waste, pyrolysis, biochar, saltwater quality

## Abstract

The pyrolysis of food waste has high economic potential and produces several value-added products, such as gas, bio-oil, and biochar. In South Korea, biochar production from food waste is prohibited, because dioxins are generated during combustion caused by the chloride ions arising from the high salt content. This study is the first to examine the water quality and the applicability of food waste-based biochar as solid refuse fuel (SRF) based on a demineralization process. The calorific value increased after demineralization due to the removal of ionic substances and the high carbon content. The chloride ion removal rate after demineralization increased with the increasing pyrolysis temperature. A proximate analysis of biochar indicated that the volatile matter decreased, while ash and fixed carbon increased, with increasing pyrolysis temperature. At 300 °C pyrolysis temperature, all domestic bio-SRF standards were met. The organic matter concentration in water decreased with increasing carbonization temperature, and the concentrations of soluble harmful substances, such as volatile organic compounds (VOCs), were within the standards or non-detectable. These results suggest that biochar can be efficiently generated from food waste while meeting the emission standards for chloride ions, dissolved VOCs, ash, and carbon.

## 1. Introduction

According to the Food and Agriculture Organization, approximately one-third of the food produced worldwide for human consumption is wasted [1,2,3]. Unfortunately, the proportion of food waste is expected to double by 2050 if proper policy changes are not implemented [4]. The wastage of food is not uniform worldwide, with industrialized countries generating more food waste than developing countries [4,5]. Central and South Asia produce the largest amounts of food waste in the world [6,7]. When food waste is decomposed, significant environmental and socioeconomic issues arise, such as greenhouse gas emissions, food insecurity, and the wastage of resources.

Based on the different properties of different food waste types, various treatment methods are employed [8]. Food waste generally comprises carbohydrates, fats, proteins, starch, hemicellulose, cellulose, and lignin, in varying amounts and compositions depending on the food waste type and source [9]. In the United States alone, more than 50 million tons of food waste are generated annually. Of these, 75% are landfilled, 18% are burned for energy recycling, and 6% are converted into compost [10]. In Singapore, most food waste is incinerated, and only 10–15% is subjected to anaerobic digestion and conversion into compost [11,12], whereas in Europe, efforts have been made to reduce food waste generation through food waste reduction programs [13,14,15]. In Korea, approximately five million tons of food waste were treated in food waste treatment facilities in 2021, approximately 77% of which is converted into feed and compost and 14% into biogas [16].

Food waste treatment methods include landfill deposition, composting through fermentation–extinction, pyrolysis to produce solid biochar and biogas, anaerobic digestion by microorganisms, conversion into animal feed, and gasification to produce synthetic gas. Recently, gasification and pyrolysis have been extensively studied for their potential for use in resource recycling [17,18].

Anaerobic digestion is the most economical among these methods, followed by heat moisture reaction, composting, incineration, and landfilling [19]. Nonetheless, in this process, a high organic loading rate under thermophilic conditions may cause instability in the digester due to the accumulation of volatile fatty acids and ammonia suppression [20].

The pyrolysis of food waste aims to produce various value-added products, such as gas, bio-oil, and biochar, by treating various feedstocks, and thus, has considerable economic potential. Additionally, it is advantageous over other treatment methods because food waste can be rapidly treated by reducing the reaction time [21]. When applied to soil, the biochar generated from food waste subjected to pyrolysis can provide water and nutrients and assist in achieving carbon neutrality [18].

Biochar obtained through food waste pyrolysis has been studied using different approaches. It has been applied to soil [9], and its potential for use in improving the adsorption capacity has been examined [22]. The potential for the use of biochar as a fuel source has also been studied [23].

However, studies on biochar production through torrefaction and its application as a fuel source are lacking compared to those on improving the biochar adsorption capacity. Particularly, the quality of saltwater generated during the demineralization process of biochar production has rarely been examined.

The problems associated with biochar production by the fuel conversion of food waste should be reviewed, with a focus on the proximate and ultimate analysis of calorific value and chlorine ion concentration associated with the combustion process. This is because a high chlorine ion concentration can release dioxins into the atmosphere from biochar burning [24], and may corrode the combustion equipment. Moreover, changes in carbon and ash contents due to increased pyrolysis temperature and demineralization can affect calorific value, combustion gas composition, and combustion equipment [25].

Food waste-based biomass, which has relatively higher sodium, potassium, and chlorine contents than coal, may increase ash deposition in boilers. Moreover, chlorine fumes from fuel during combustion form alkali chlorides, such as NaCl and KCl, by accelerating the vaporization of alkali metals, and may increase iron corrosion by chlorides and ash deposition in the boiler [26,27].

Herein, food waste-based biochar was examined according to domestic bio-soil refuse fuel (SRF) standards and European BS EN standards for its utility as a fuel source. Problems occurring during biochar combustion were addressed through demineralization, and the quality of the saltwater generated through demineralization was also examined.

## 2. Result and Discussion

### 2.1. Production Yield of Food Waste-Based Biochar

Table 1 shows the production yield of food waste-based biochar according to temperature and residence time. The yield was calculated based on a 10 kg of dry food waste input. As the temperature and residence time increased, the biochar amount and yield decreased. The biochar yield significantly varies depending on the properties of raw materials. The maximum biochar yield of 43.16% has been reported during the pyrolysis of peanut shells at 350 °C [28], while the yields of food waste-based biochar at 300, 400, 500, and 700 °C have been reported as 60.55, 42.61, 38.05, and 35.03%, respectively [29]. The yield acquired in this study was higher because the residence time (10 and 30 min) was shorter than that employed previously (4 h). For biochar production, at a low pyrolysis temperature of less than 300 °C, a large amount of energy is used to evaporate moisture, and the applicability of the produced biochar as fuel decreases due to the low carbon content [30]. According to a previous study [31], the biochar and energy yields according to the temperature are similar. Since the biochar yield was high at 300 °C in the present study, the energy yield was expected to be high.

### 2.2. Heating Value and Chloride Ions of Food Waste-Based Biochar before and after Demineralization

Figure 1 shows the higher heating value (HHV) before and after demineralization according to temperature and residence time. The HHV after demineralization was higher than that before demineralization in all temperature ranges. At 300, 400, and 500 °C, the difference in HHV before and after demineralization was 1.1 and 0.1 MJ/kg, 0.4 and 1.2 MJ/kg, and 1.0 and 0.8 MJ/kg for a residence time of 10 and 30 min, respectively. The HHV of the raw sample was 18.7 MJ/k and that of food waste-based biochar was 21.0–24.5 MJ/kg. Thus, the calorific value of the biochar was similar to that of coal (26–28 MJ/kg) [32].

In all temperature ranges, the HHV of food waste-based biochar increased after demineralization (Figure 1), while the chlorine concentration decreased after demineralization. HHV increases with temperature and residence time because ionic substances are released inside the biochar, and there is an increase in carbon content [33,34]. The HHV increased at 400 °C because this temperature triggered the pyrolysis reaction; subsequently, it slightly decreased at 500 °C [31]. The increase in HHV after demineralization appeared to be due to the removal of salt, as well as alkali and alkaline earth metals, in the biochar. These components are discharged from food waste during demineralization, and the calorific value increases due to the increase in carbon content compared to the total weight [35,36].

The domestic standard of bio-SRF [37] states a heating value of 3000 kcal/kg (12.56 MJ/kg). This was met by the raw sample at all temperature ranges. In the standard of BS EN 15359:2011 [38], the net calorific values are 25, 20, 15, 10, and 3 MJ/kg or higher for Grades 1–5, respectively. In our analyses, all samples showed a calorific value corresponding to Grade 2 or higher before and after demineralization.

When food waste-based biochar is converted into fuel, the salt concentration in various components of biochar generates dioxins [39], and large amounts of salt impede the conversion. Figure 2 shows the changes in the chlorine concentration of food waste-based biochar. After biochar formation, the chlorine concentration before demineralization increased compared to that in the raw sample. The chlorine concentration after demineralization decreased compared to that before demineralization for all samples. Moreover, the chlorine concentration reduced significantly when the residence time was 30 min at 500 °C, but overall, it increased with temperature. The salt removal efficiency was higher for 30 min than for 10 min. The differences in the removed chlorine ion concentrations between before and after demineralization for 10 min and 30 min were 1.3% and 1.5% at 300 °C, 1.5% and 1.9% at 400 °C, and 1.8% and 2.0% at 500 °C, respectively. This indicates that the amount of chloride ions removed by demineralization was high at high carbonization temperatures and long residence times. While many substances are volatilized into gas during pyrolysis, salt retains its crystalline structure without being volatilized, and is removed during demineralization as the crystals dissolve [24].

When the salt in food waste is produced as biochar, Na and Cl exist together rather than as separate ions, and they are not volatilized during pyrolysis. Moreover, salt in food waste remains crystalline during biochar formation, which makes it possible to demineralize the crystalized biochar salt with water [35], and further dehydrate the dissolved salt by centrifugation.

In this study, the removal efficiency of chloride ions through demineralization was 58.4–91.2%.

### 2.3. Proximate and Ultimate Analyses of Biochar before and after Demineralization at Different Temperatures

Figure 3 shows that as the temperature increased, volatile matter decreased, while ash and fixed carbon increased. This illustrates the intensification in devolatilization and pyrolysis reactions as the temperature increased. The increase in ash content at high temperatures can hinder fuel conversion [40,41,42]. In a thermochemical reaction, biomass-based (e.g., wood composites) biochar emits various types of volatile substances owing to the decomposition of biopolymers (cellulose, hemicellulose, and lignin) [43]. Hemicellulose and cellulose are decomposed at 200–400 °C, followed by the decomposition of lignin at 400–650 °C [44]. Food waste includes carbohydrates, proteins, and fats. Of these, proteins and carbohydrates are pyrolyzed at 150–360 °C, while fats are pyrolyzed at 330–560 °C [45]. The contents of these volatile substances decrease with increasing pyrolysis temperature, indicating that the residual organic matter in biochar is slowly discharged by pyrolysis. Fixed carbon increases the carbon conversion rate at high temperature and improves the carbonization level of biochar through organic matter decomposition [46]. Herein, at all temperature ranges, the volatile matter and fixed carbon increased while ash decreased after demineralization, indicating that demineralization can assist in fuel conversion.

Figure 4 shows that as the temperature increased, carbon and nitrogen increased, while hydrogen and oxygen decreased. This was due to the influence of the dehydration and decarbonization occurring during the carbonization of food waste [25,47].

Further, the sulfur content was significantly lower than that of other elements and decreased with increasing temperature. Carbon and fixed carbon increased after demineralization. Typical wood biochar has low chlorine, nitrogen, and ash contents, but agricultural materials, such as straw, contain chlorine and alkali metals, such as potassium and sodium. Wood-based biomass fuel generally has a low ash content and extremely low nitrogen and sulfur contents, compared to those in coal [32]. In the case of food waste-based biochar, however, the nitrogen content was high, with little or no sulfur, owing to the presence of protein.

### 2.4. Component Analysis According to the Bio-SRF Standard

Table 2 shows the component analysis results of biochar following the domestic bio-SRF standard in accordance with the “Enforcement Decree of the Act on the Promotion of Saving and Recycling of Resources” [37,48]. A comparative analysis was conducted according to the non-pellet standard. Under all temperature conditions, the bio-SRF standard was met. While ash content increased with temperature, it remained within the bio-SRF thresholds before and after demineralization at 300 °C. Based on the Wastes Control Act of Korea (2020), the SRF standard was applied to solid fuel produced from municipal waste, but an attempt was made to apply the stricter standard of bio-SRF. All assessed components, except for ash and chloride ions, were within the bio-SRF standard before and after demineralization. Particularly, ash content was within the standard at 300 °C, and chloride ions were within the standard after demineralization. The ash content of biochar depends on the properties and temperature of the raw material, and increases with carbonization temperature [49,50]. The biochar carbonized at 300 °C increases the energy content by approximately 30% compared to unprocessed biomass [51]. Since an increase in the moisture and ash content of biochar decreased the energy content of biomass components, moisture and ash contents should be properly maintained to meet the combustion standards of food waste-based biochar.

Chlorine was within the standard after demineralization. All items, except for chloride ions, met the bio-SRF standard in all temperature ranges. BS EN 15359:2011 classifies fuels into five grades based on calories as well as chlorine and mercury contents. For chlorine contents, this corresponds to a median of 0.2%, 0.6%, 1.0%, 1.5%, and 3% or less for Grades 1 to 5, respectively [38,52]. In our results, the chlorine content was of Grade 2 or less at all temperatures after demineralization. Mercury content, based on the classification of median values of 0.02, 0.03, 0.08, 0.15, and 0.5 mg/MJ for Grades 1 to 5, respectively, was within the first grade in all cases. Among the heavy metals, the chromium content was relatively higher than that of other components, showing similar concentrations to that in sewage sludge [53]. Among the food waste components, heavy metals have various sources [54]. Typical chromium species include trivalent and hexavalent chromium with underwater stability. Trivalent chromium is involved in the metabolism of carbohydrates, fats, and proteins, while hexavalent chromium is a toxin and carcinogen [55,56]. Hexavalent chromium has not been previously observed in food waste-based biochar. In the present study [57], most chromium was judged to constitute trivalent chromium. The food waste-based biochar, which reduced the dependence on coal and utilized biomass as fuel, met all conditions for the bio-SRF standard before and after demineralization at 300 °C. To meet the bio-SRF standards at temperatures above 300 °C, further research on ash reduction is required.

### 2.5. Saltwater Quality Characteristics

Table 3 shows the water quality analysis results for the saltwater according to the temperature. In Korea, emission standards are classified into clean areas, A, B, and special areas, for each water pollutant discharged from wastewater discharge facilities, according to the emission standards of water pollutants in the “Enforcement Rules of The Water Environment Conservation Act (2021)” [58]. The emission standards of water pollutants are organic matter emission standards, which include BOD, TOC, SS, and 52 other items, including hydrogen ion concentration and water pollutants, such as phenols. Of these, we focused on analyzing underwater VOCs because the volatile matter in food waste-based biochar can be included in the saltwater during demineralization. The higher pyrolysis temperature of biochar resulted in a lower organic matter content in the water. VOCs were not eluted from the biochar, except formaldehyde, the concentration of which was extremely low and within the standard of a clean area. Higher pyrolysis temperatures were correlated with higher chloride ion concentrations and lower sulfate and phosphate ion concentrations in the water. When the quality of saltwater produced before and after biochar demineralization was analyzed, the organic matter concentration was found to be high at a low temperature of 300 °C, but was low at high temperatures. Indicators of organic matter (BOD, TOC, TN, TP, and SS) decreased as the carbonization temperature increased, which was possibly because of the volatilization of organic matter in large quantities during the carbonization process with increasing temperatures. Further, the organic matter concentrations were significantly lower than the average concentrations of the leachate from domestic food waste (BOD, TN, TP and SS: 57,870, 5280, 880 and 160,000 mg/L, respectively) [59]; nevertheless, they did not meet the emission standards for clean areas. Since the organic matter concentrations exceeded the regulation amounts, organic matter levels should be reduced before discharge. Moreover, as VOCs include artificially synthesized compounds, Environment Protection Act standards are strict for drinking water [60]. In our tests, dissolved pollutants, such as VOCs, were within the standard or undetectable, and thus, water pollution by VOCs was not an issue. A larger amount of chloride ions was discharged as the carbonization temperature increased because food waste-based biochar was demineralized with a crystalline structure [30]. A comparison of the underwater Cl concentration (Table 3) with the Cl concentration in biochar before and after demineralization (Figure 2) showed that the difference in concentration increased at high temperatures for biochar and saltwater before and after demineralization. In the latter, the dissolved amounts of sulfate and phosphate ions decreased with increasing temperature. The dissolved VOCs, which were generated through the demineralization of biochar, were within the emission standard, while those in the saltwater of food waste-based biochar had concentrations similar to those of the inflow water (sewage) entering sewage treatment plants [61]. We therefore suggest ash content and chloride ions should be carefully controlled during the conversion of food waste biochar into fuel according to the SRF standard.

## 3. Materials and Methods

### 3.1. Materials

Most of the food waste generated in the Seoul metropolitan area of Korea is treated by drying, feed conversion, and composting processes. In our study, a sample of dry food waste was used to evaluate the energy efficiency of biochar produced from food waste. The sample comprised representative household food waste, and was produced using the input hopper, crusher, measuring tank, drier, and magnetic separator in the Gimpo Resource Recycling Center. No additional energy was used to dry the food. The center dries approximately 30 tons of the food waste generated in Gimpo City and produces 3–4 tons of dry feed daily. Table 4 shows the ultimate and proximate analysis results of the dry sample.

### 3.2. Kiln Description

#### 3.2.1. Kiln

The kiln (rotary kiln; Kiln & Furnace Tech, Ansan-si, Korea) used in the experiment was an electric model with a screw-type interior that circulated and carbonized the added food waste sample. The sample was injected using the feeder at the sample inlet. Nitrogen gas was injected during the initial stage to allow operation under anoxic conditions. The maximum internal temperature of the carbonization system was 600 °C and the heating rate was set to 8 °C/min. The residence time of food waste could be adjusted to 10–40 min. The input amount of the sample was adjusted according to the residence time, with a maximum amount of 20 kg/h. We used 10 kg of dry sample for each combination of temperature (300, 400, and 500 °C) and residence time (10 and 30 min). Figure 5 shows the pyrolysis and combustion system flow chart for biochar production.

#### 3.2.2. Demineralization System

The demineralization process applied after biochar production from food waste consisted of mixing with water through agitation, followed by demineralization through the separation of water and biochar by centrifugation. The weight ratio between biochar and water was set to 1:10. The separated biochar was filtered by centrifugation at approximately 600 rpm in a filter-type centrifugal dehydrator equipped with a 200 mm mesh filter.

The combustion system was designed to maintain an average temperature of 400 °C to prevent high-concentration odors.

### 3.3. Sample Analysis

The heating value of the food waste-based biochar was measured using a bomb calorimeter (6400 Automatic Isoperibol Calorimeter, Parr, Moline, IL, USA). For chlorine measurement, an ion chromatograph (AQF-2100H, Mitsubishi Chemical Analytech Co., Ltd., Kanagawa, Japan) was used. Proximate analysis was conducted according to the American Society for Testing and Materials D7582 experimental standard. For ultimate analysis, 2400 series II CHNS/O (Perkin Elmer, Boston, MA, USA) was used. The heavy metals in bio-SRF were analyzed using a mercury analyzer (M7600, Teledyne, Thousand Oaks, CA, USA), and an inductively coupled plasma-optical emission spectrometer (Agilent 720, Agilent, Santa Clara, CA, USA) was used to analyze ionic components. To analyze saltwater quality, biochemical oxygen demand (BOD), suspended solids (SS), total nitrogen (TN), and total phosphorus (TP) were analyzed using a standard method [62]. Total organic carbon (TOC) was analyzed using a TOC-L analyzer (Shimadzu, Kyoto, Japan). Chloride, sulfate, phosphate, nitrate, nitrite, and bromide ions were analyzed using an ion chromatograph (ICS-1100, Thermo Fisher Scientific, Waltham, MA, USA). The remaining volatile organic compounds (VOCs) were analyzed using a high-sensitivity (HS)–gas chromatography/mass spectrometer (QP2020NX, Shimadzu, Kyoto, Japan).

### 3.4. Experimental Method

The dry food waste sample was subjected to pyrolysis at 300, 400, and 500 °C for 10 and 30 min. After biochar production, each sample was subjected to demineralization by agitating it in a 200 L reaction tank with an agitator, and water was separated from the biochar using a centrifuge. Sampling for each configuration was performed more than three times to ensure analysis accuracy, and the statistical analysis was conducted using Microsoft Excel 2019.

## 4. Conclusions

Herein, biochar was generated at carbonization temperatures of 300, 400, and 500 °C and residence times of 10 and 30 min to examine the conversion of food waste into fuel. Additionally, biochar characteristics before and after demineralization were examined. The production yield of biochar decreased with increasing temperature and residence time. The calorific value was higher after than before demineralization because of the removal of ionic substances and the increased carbon content. The concentration of chloride ions in biochar increased with the pyrolysis temperature, and a higher pyrolysis temperature was correlated with greater chlorine ion removal after demineralization. The proximate analysis of biochar after demineralization by temperature revealed that volatile components decreased, and ash and fixed carbon increased, with temperature. The results met the bio-SRF standard at all temperatures. While ash content increased with temperature, the standard was met before and after demineralization at 300 °C. An examination of the quality of the produced demineralized water shows that the concentration of organic matter in the water produced after demineralization decreased as the carbonization temperature increased. Concentrations of soluble harmful substances, such as VOCs, were within the standard or non-detectable. Higher pyrolysis temperatures resulted in higher concentrations of chloride ions eluted in the water, and lower concentrations of ionic substances, such as sulfate and phosphate ions. Overall, the biochar produced through the pyrolysis of food waste was confirmed as a suitable option for fuel generation.

## Figures and Tables

**Figure 1 molecules-28-06114-f001:**
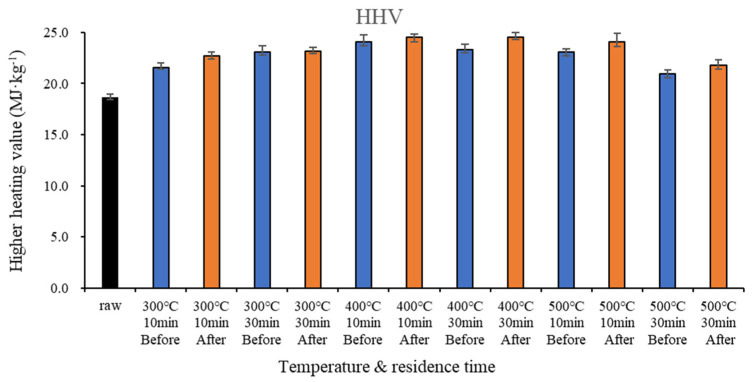
HHV of biochar by temperature before and after demineralization.

**Figure 2 molecules-28-06114-f002:**
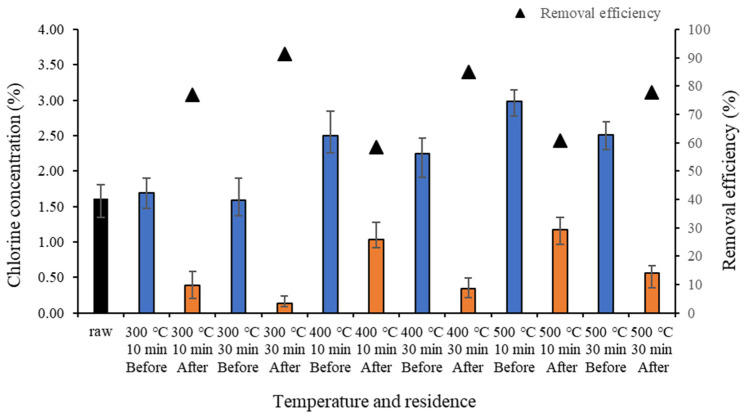
Chlorine concentration and removal efficiency of biochar by temperature before and after demineralization.

**Figure 3 molecules-28-06114-f003:**
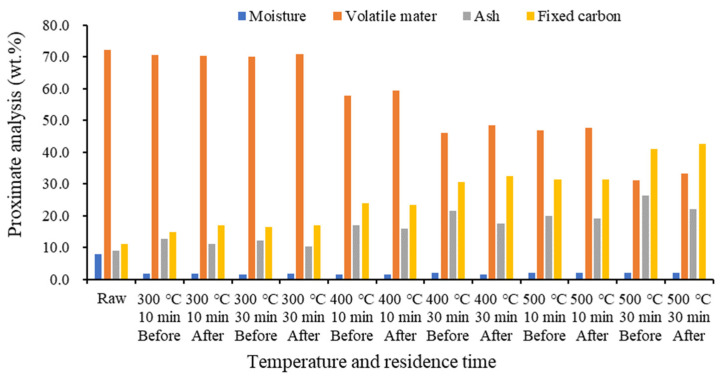
Proximate analysis results of biochar by temperature before and after demineralization.

**Figure 4 molecules-28-06114-f004:**
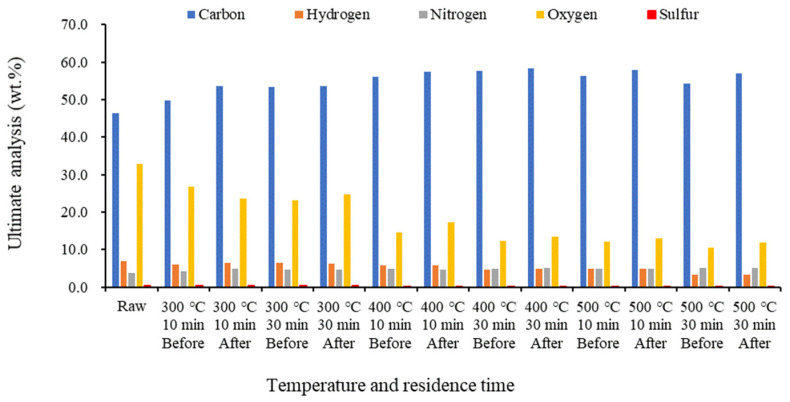
Ultimate analysis results of biochar by temperature before and after demineralization.

**Figure 5 molecules-28-06114-f005:**
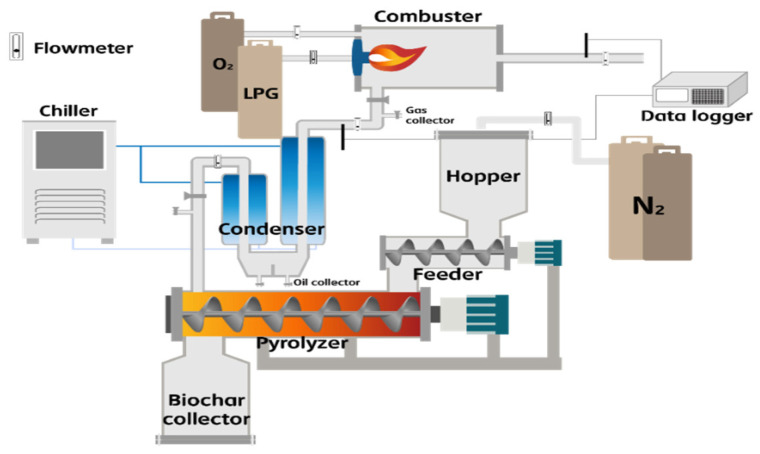
Schematic diagram of pyrolysis device used for biochar production.

**Table 1 molecules-28-06114-t001:** Production yield of food waste-based biochar by temperature and residence time.

Temperature (°C), Residence Time (min)	Before Pyrolysis (kg)	After Pyrolysis (kg)	Yield (%)
300, 10	10.0 ± 0.4	7.66 ± 0.4	76.6
300, 30	10.0 ± 0.3	7.22 ± 0.2	72.2
400, 10	10.0 ± 0.4	5.87 ± 0.5	58.7
400, 30	10.0 ± 0.2	5.46 ± 0.4	54.6
500, 10	10.0 ± 0.4	5.38 ± 0.3	53.8
500, 30	10.0 ± 0.4	4.32 ± 0.4	43.2

**Table 2 molecules-28-06114-t002:** Bio-SRF component contents of biochar.

Component	Unit	300 °C 30 Min before	300 °C 30 Min after	400 °C 30 Min before	400 °C 30 Min after	500 °C 30 Min before	500 °C 30 Min after	Bio-SRF (Non-Pellet)	SRF (Non-Pellet)
Moisture	wt.%	1.54	1.84	2.00	1.53	1.96	2.05	≤25	≤25
Ash	wt.%	12.14	10.33	21.35	17.6	26.25	22.1	≤15	≤20
Chlorine	wt.%	1.59	0.14	2.24	0.34	2.51	0.56	≤0.5	≤2.0
Sulfur	wt.%	0.23	0.21	0.16	0.2	0.12	0.2	≤0.6	≤0.6
Biomass	wt.%	>99.5	>99.5	>99.5	>99.5	>99.5	>99.5	≥95	
Mercury	(mg·kg^−1^)	<0.01	<0.01	<0.01	<0.01	<0.01	<0.01	≤0.6	≤1.0
Cadmium	(mg·kg^−1^)	0.45	0.51	0.62	0.75	0.48	0.53	≤5.0	≤5.0
Lead	(mg·kg^−1^)	<1.5	<1.5	<1.5	<1.5	<1.5	<1.5	≤100	≤150.0
Arsenic	(mg·kg^−1^)	0.76	0.37	0.32	0.22	0.35	0.31	≤5.0	≤13.0
Chromium	(mg·kg^−1^)	56.1	48.3	15.8	20.5	35.2	44.1	≤70.0	

**Table 3 molecules-28-06114-t003:** Water quality analysis results for salt water (N.D., not detected; VOCs, volatile organic carbons).

Analysis Item		Unit	300 °C 30 Min	400 °C 30 Min	500 °C 30 Min	Clean Area Standard
Organic matter	Biochemical oxygen demand	mg/L	999.00	141.00	118.00	≤30
	Total organic carbon	mg/L	859.00	64.20	43.20	≤25
	Suspended solids	mg/L	75.00	23.50	11.60	≤30
	Total nitrogen	mg/L	69.40	13.00	8.83	≤30
	Total phosphorus	mg/L	13.40	1.92	0.73	≤4.0
VOCs	Fluorine	mg/L	ND	ND	ND	≤3
	Trichloroethylene	mg/L	ND	ND	ND	≤0.06
	Tetrachloroethylene	mg/L	ND	ND	ND	≤0.02
	Dichloromethane	mg/L	ND	ND	ND	≤0.02
	Benzene	mg/L	ND	ND	ND	≤0.01
	Carbon tetrachloride	mg/L	ND	ND	ND	≤0.004
	1,1-dichloroethylene	mg/L	ND	ND	ND	≤0.03
	1,2-dichloroethane	mg/L	ND	ND	ND	≤0.03
	Chloroform	mg/L	ND	ND	ND	≤0.08
	1,4-dioxane	mg/L	ND	ND	ND	≤0.05
	Vinyl chloride	mg/L	ND	ND	ND	≤0.01
	Acrylonitrile	mg/L	ND	ND	ND	≤0.01
	Bromoform	mg/L	ND	ND	ND	≤0.03
	Naphthalene	mg/L	ND	ND	ND	≤0.05
	Formaldehyde	mg/L	0.10	0.03	0.04	≤0.5
	Toluene	mg/L	ND	ND	ND	≤0.7
	Xylene	mg/L	ND	ND	ND	≤0.5
	Styrene	mg/L	ND	ND	ND	≤0.02
Ionic substances	Chloride ion, Cl^−^	mg/L	374.00	422.00	621.00	-
	Sulfate ion, SO_4_^2−^	mg/L	44.50	16.10	14.50	-
	Phosphate ion, PO_4_^3−^	mg/L	33.40	4.70	1.70	

**Table 4 molecules-28-06114-t004:** Ultimate and proximate analyses of the dry food waste sample used in the experiment (HHV, high heating value).

Proximate Analysis (wt. %)				Ultimate Analysis (wt. %)						Others	HHV (MJ/kg)
Moisture	Volatile Matter	Ash	Fixed Carbon	C	H	O	N	S	Cl		
7.80	72.21	8.89	11.10	46.43	6.92	3.18	32.89	0.31	1.61	6.16	18.7

## Data Availability

Not applicable.

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
