# Peer review of "Biochar Production and Demineralization Characteristics of Food Waste for Fuel Conversion"

_molecules, 2023, doi:10.3390/molecules28166114_

Round 1

Reviewer 1 Report

More English revisions 

Author Response

Response to Reviewer 1 Comments

Point 1: Comments and Suggestions for Authors : More English revisions

Response 1: Thank you for your comments. We have reviewed the document and revised the text throughout for English language quality.

Reviewer 2 Report

Journal: Molecules

Ms. ID.: molecules-2513433

Title: Biochar Production and Demineralization Characteristics of Food Waste for Fuel Conversion

Ahn et al. aim to generate biochar through the pyrolysis of food waste. They used chromatography, spectrometry, bomb calorimeters, and HS-GC/MS to examine biochar components, as well as other components obtained during the process of pyrolysis. The topic is interesting, actual, and in the scope of the journal. Still, the manuscript needs some serious improvements in order to be published. My comments are listed below.

-The abstract needs to be completely rewritten since it is confusing. Many sentences are too long and should be divided for a better understanding of the text.

-The Introduction is informative in general but should provide further clarification on why it is important to examine all components obtained during the process of pyrolysis.

-The authors should specify what exactly the food waste they used is.

-The results are presented in an acceptable format, but the discussion needs improvement. The conclusions are superficial and confusing. Many of these issues are related to the language barrier probably.

-lines 180-181: “Among the components of food waste, heavy metals have various sources. Chrome is known to be generated even from plastic [43].” – this is not relevant since plastics are not food waste.

-The conclusion is missing, and it would provide a critical summarization of somewhat confusing results. 

Extensive editing is required. 

Author Response

Response to Reviewer 2 Comments

Ahn et al. aim to generate biochar through the pyrolysis of food waste. They used chromatography, spectrometry, bomb calorimeters, and HS-GC/MS to examine biochar components, as well as other components obtained during the process of pyrolysis. The topic is interesting, actual, and in the scope of the journal. Still, the manuscript needs some serious improvements in order to be published. My comments are listed below.

Point 1: The abstract needs to be completely rewritten since it is confusing. Many sentences are too long and should be divided for a better understanding of the text.

Response 1: Thank you for your comments. We have modified the Abstract section accordingly.

Point 2: The Introduction is informative in general but should provide further clarification on why it is important to examine all components obtained during the process of pyrolysis.

Response 2: We have revised the Introduction accordingly, and inserted some additional material (line 73~80) to clarify the importance of this investigation.  

Point 3: The authors should specify what exactly the food waste they used is.

Response 3: We have inserted some clarifications about the type of food waste used in section 4.1 (Materials) (line 244 following).

Point 4: The results are presented in an acceptable format, but the discussion needs improvement. The conclusions are superficial and confusing. Many of these issues are related to the language barrier probably.

Response 4: Thank you for your constructive comments. We have revised the structure and content of the Discussion section and substantially expanded the Conclusion section (line 301 following).

Point 5: lines 180-181: “Among the components of food waste, heavy metals have various sources. Chrome is known to be generated even from plastic [43].” – this is not relevant since plastics are not food waste. lines 180-181:

Response 5: Thank you for pointing this out; we have removed the irrelevant sentences.

Point 6: The conclusion is missing, and it would provide a critical summarization of somewhat confusing results.

Response 6: Thank you for your comments.  We have inserted a substantially expanded conclusion section (line 301).

Reviewer 3 Report

1. Introduction. -please add some information about typical food waste composition

Inn overall the structure of the article must to be reorganized: Materials, Experimental Results, Discussion,  Summary. In this form is difficult to read. Discussion should be improved. 

Line 153: As mentioned previously, the heating value after demineralization was higher than that... Please add some detailed information. 

Line 217: Please add sample detailed composition

Line 232: The mass of the sample was 20 kg in each experiment? 

The obtained results should be compare with EBC standards.

In overall manuscript presents interesting results. however, it should include more scientific discussion, a comparison with e.g. hydrocabonization. In this form it is a scientific report. 

In general, the manuscript is written legibly

Author Response

Response to Reviewer 3 Comments

Point 1: 1. Introduction. -please add some information about typical food waste composition

Response 1: Thank you for your comment. We have added some additional information on common food waste composition to the Introduction (line 40-42).

Point 2: Inn overall the structure of the article must to be reorganized: Materials, Experimental Results, Discussion,  Summary. In this form is difficult to read. Discussion should be improved.

Response 2: Please note that we have arranged the sections according to the journal’s preferred order (which requires Results and Discussion to precede the Materials and Methods). We have revised the Discussion section (line 172 following) and added a substantially expanded Conclusion section (line 301 following).

Point 3: Line 153: As mentioned previously, the heating value after demineralization was higher than that... Please add some detailed information.

Response 3: We have rephrased these sentences to clarify the intended comparison (earlier in this paragraph, and Figure1).

Point 4: Line 217: Please add sample detailed composition

Response 4: We have added information about sample composition in section 4.1 (line 244 following) and in Table 4.

Point 5: Line 232: The mass of the sample was 20 kg in each experiment?

Response 5: While the maximum input to the pyrolysis unit per hour was 20 kg, the sample weight for each experiment was 10 kg. This has been clarified in section 2.1 (line 87).

Point 6: The obtained results should be compare with EBC standards.

Response 6: Thank you for your suggestion. The European Biochar Certificate (EBC) is a regulation for agricultural biochar, intended to be applied to lignocellulosic biochar used as a soil improver or fertilizer (https://www.european-biochar.org/en/). Since this study rather focused on the use of food waste-based biochar as fuel, we used Bio-SRF and BS EN as the applicable standards, and also reviewed the environmental aspects of the produced demineralized water.

Point 7: In overall manuscript presents interesting results. however, it should include more scientific discussion, a comparison with e.g. hydrocabonization. In this form it is a scientific report.

Response 7: Thank you for your comment. HTC and pyrolysis involve different production methods for biochar. HTC is carried out at temperatures between 180 °C and 250 °C in a closed system, using water as the reaction medium, while we focused on food waste dried samples that were thermally decomposed at 300 to 500 degrees (the pyrolysis reaction according to the temperature is covered in the Discussion). Comparisons of the two methodologies have already been carried out (e.g., Correa et al. (2019)), and we suggest that this should not be a focus of our present paper.

Reference:

Correa, C. R., Hehr, T., Voglhuber-Slavinsky, A., Rauscher, Y., & Kruse, A. (2019). Pyrolysis vs. hydrothermal carbonization: Understanding the effect of biomass structural components and inorganic compounds on the char properties. Journal of Analytical and Applied Pyrolysis, 140, 137-147.

Round 2

Reviewer 2 Report

The authors have made some revisions to the manuscript, but regrettably, these changes appear to be insufficient. The abstract, introduction, and discussion sections have only undergone minimal alterations. I strongly recommend that the authors make further efforts to improve the text, ensuring its quality and making it suitable for publication in a reputable scientific journal. 

Moderate changes are required. 

Author Response

Response to Reviewer 2 Comments

Point 1: The authors have made some revisions to the manuscript, but regrettably, these changes appear to be insufficient. The abstract, introduction, and discussion sections have only undergone minimal alterations. I strongly recommend that the authors make further efforts to improve the text, ensuring its quality and making it suitable for publication in a reputable scientific journal.

Response 1: Thank you for your comments. According to the reviewer’s comments, we have further corrected the manuscript for language, readability, and grammar. The flow of ideas has also been improved at several instances (e.g., in the Materials and Method s section) by re-arranging the content within paragraphs. Moreover, repetitive content has been deleted to ensure conciseness in the text. We thoroughly reviewed the document and revised it by supplementing suitable scientific evidence.

Reviewer 3 Report

Usually Materials, Methods... comes first. After improving the organization of the section, the manuscript can be published.

Author Response

Response to Reviewer 3 Comments

Point 1: Usually Materials, Methods… comes first. After improving the organization of the section, the manuscript can be published.

Response 1: According to your comments, we have reorganized the paper in the order of introduction, materials and methods, result and discussion, and conclusion. Correspondingly, the citation numbers and figure/table numbers have been corrected in the captions as well as in the citations in the text to follow the new order of sections.